# Potential Effects of Human Papillomavirus Type Substitution, Superinfection Exclusion and Latency on the Efficacy of the Current L1 Prophylactic Vaccines

**DOI:** 10.3390/v13010022

**Published:** 2020-12-24

**Authors:** Ian N. Hampson, Anthony W. Oliver, Lynne Hampson

**Affiliations:** Division of Cancer Sciences, University of Manchester Viral Oncology Labs, Research Floor, St Marys Hospital, Oxford Rd, Manchester M13 9WL, UK; anthony.w.oliver@manchester.ac.uk (A.W.O.); lynne.hampson@manchester.ac.uk (L.H.)

**Keywords:** HPV, CIN, prophylactic vaccine, L1, superinfection exclusion, type replacement, latency

## Abstract

There are >200 different types of human papilloma virus (HPV) of which >51 infect genital epithelium, with ~14 of these classed as high-risk being more commonly associated with cervical cancer. During development of the disease, high-risk types have an increased tendency to develop a truncated non-replicative life cycle, whereas low-risk, non-cancer-associated HPV types are either asymptomatic or cause benign lesions completing their full replicative life cycle. HPVs can also be present as non-replicative so-called “latent” infections and they can also show superinfection exclusion, where cells with pre-existing infections with one type cannot be infected with a different HPV type. Thus, the HPV repertoire and replication status present in an individual can form a complex dynamic meta-community which changes with respect to both time and exposure to different HPV types. In light of these considerations, it is not clear how current prophylactic HPV vaccines will affect this system and the potential for iatrogenic outcomes is discussed in light of recent outcome data.

## 1. Introduction

Although published work overwhelmingly supports the efficacy of the current prophylactic HPV vaccines, there has been a great deal of publicity and numerous studies carried out on potential vaccine-related adverse events [1,2,3,4]. The general consensus is that there is no strong evidence that these are increased when compared to other similar adjuvant-containing products although this is still a controversial subject since there are some reports which claim increased adverse reactions in relation to the HPV vaccines [2,5]. However, it is not intended to discuss this aspect of the vaccines further. Instead, the main focus will be the complex biology of mucosal HPVs and their potential for interacting with the vaccines to produce unexpected outcomes.

## 2. HPV Types and Associated Pathologies

Although there are ~200 distinct types of HPV, there is substantial evidence that many others are yet to be identified [6,7] as exemplified by the recent identification of a novel Beta-2 papillomavirus from skin [8]. Significantly, these have a range of different tissue tropisms and disease associations (Table 1) and there are 51 distinct HPV types known to infect genital mucosal epithelium [9].

Of these, ~14 are classed as high risk and are associated with cervical cancer although some are more carcinogenic than others [6,9,10]. For example, type 16 has the highest oncogenic potential and is usually followed by type 18, whereas low-risk HPV 6 and 11 are more commonly associated with genital warts. It is also notable that HPV also causes tumours at other anatomical sites such as the anus, vulva, penis and oropharynx [11,12,13]. In addition to the 14 high-risk types, there are 6 possibly high-risk and 31 low-risk HPV types found in genital epithelium although there is a distinct paucity of information on the incidence of the latter in both normal and abnormal cervical epithelium. In order to address this issue Schmitt et al. analysed the prevalence of 51 HPV types in 1273 smears obtained from Belgian women and correlated the results with normal versus abnormal cytology [9]. As expected, there was an increase in the prevalence of high-risk types associated with increasing grade of abnormal cytology although infection with low-risk types was far more common than anticipated. Most significantly, only women with normal cytology showed a higher viral load for low-risk HPVs than for high-risk types present, whereas this was reduced to one third the level of high-risk types found in women with high-grade squamous intraepithelial lesions (HSIL). Curiously low-risk HPVs, such as type 42, showed a >5-fold increase from normal to low-grade squamous intraepithelial lesions (LSIL) but then dropped to half this level in HSILs. These data indicate that development of HSIL is associated with an increase in the viral load of high-risk HPVs but could also be associated with a concomitant decrease in the viral load of some low-risk types. In this regard, low-risk HPV42 was found to be the most common HPV type detected in cervical smears collected from women in the Apulia region of Italy although it is acknowledged that there are significant variations in HPV types present with respect to different geographical locations [14].

## 3. Normal versus Truncated HPV Life Cycles and the Development of Neoplasia

The unique life cycle of HPV is intimately associated with how the virus manages to evade the immune system. At its simplest, the virus hides by not penetrating below the basement membrane where the majority of immune effector cells reside. Furthermore, it does not produce secreted proteins, inflammation, viremia or cell death [7,15]. There are eight viral gene products and the L1 protein is produced late in the virus life cycle. L1 is a capsid protein component of mature virions which are only produced in post-mitotic, superficial cells of the epidermis in order to avoid detection by the host immune system (Figure 1 obtained from [16]) [7]. In the normal HPV life cycle, the virus exists as an extrachromosomal episome where the active infection is maintained by expression of the early (E) proteins which are expressed at low levels in basal cells of the epithelium.

However, during the development of HPV-related high-grade cervical neoplasia, the production of mature virions is reduced since the virus downregulates its full replicative life cycle and expression of the L1 protein is also reduced (Figure 2 obtained from [17]). This transition from episomal virus replication most often occurs as a result of loss of function of the E2 protein by disruption of the E2 open reading frame (ORF) when the virus integrates with cellular DNA of the host as illustrated in Figure 2 [18,19,20,21]. Alternatively, the function of the intact E2 protein can also be suppressed by altered methylation of its binding sites in the viral LCR [22]. Indeed, epigenetic gene regulation is known to play a crucial role in HPV-related neoplastic progression, whereby methylation of the virus LCR, in addition to L1, L2 and E5 ORFs, has been evaluated as a marker of disease [23,24,25]. Since the E2 protein negatively regulates transcription of the HPV E6 and E7 oncogenes, reduced E2 function significantly upregulates their expression [18,19] which, in turn, promotes genetic instability combined with many other pro-oncogenic effects [26]. Furthermore, reduced E2 function also down-regulates expression of the splicing factor SRSF3 which subsequently leads to down-regulated expression of L1 [27]. Most significantly, ectopic expression of different E2 proteins with low sequence homology (40–56% AA Identity) from HPVs 52, 53 and 58 in cervical cancer cells containing integrated HPV18 (HeLa) and HPV16 (CaSki), demonstrated they all possessed the promiscuous ability to upregulate L1 expression from integrated endogenous virus in these cells [28].

## 4. Mode of Action of L1 Targeted Prophylactic Vaccines

Cervarix targets the L1 protein from HPV 16/18 (bivalent), Gardasil targets L1 from HPV16/18/6/11 (quadrivalent) and Gardasil 9 targets L1 from 16/18/31/33/45/52/58/6/11 (nonavalent) and their prophylactic use is currently advocated for women prior to infection with HPV. They all induce HPV type-specific L1 antibodies which provide humoral immunity against HPV infections in HPV-negative women by blocking the virus from interacting with its cellular receptors and many studies have demonstrated their effectiveness at protecting against CIN lesions caused by infection with vaccine-covered HPV types. However, for >15 years, there has been considerable debate on the ability of Cervarix and Gardasil to protect both HPV-positive and -negative women from HPV-associated cervical cancer. For example, it is well known that HPV-negative women vaccinated with either the bivalent or quadrivalent vaccines can still develop CIN lesions positive for HPV types not covered by these products although it is anticipated that the more recently introduced Gardasil 9 should improve this [29,30]. Furthermore, there is also evidence of some cross-protection against other non-vaccine high-risk types for both Cervarix and Gardasil [31,32,33].

In women who are HPV positive at the time of vaccination, additional concerns are based on the mode of action of the L1 targeted vaccines. Since expression of the L1 protein is downregulated as a consequence of the truncated HPV life cycle that is a common feature of high-grade type 3 cervical intraepithelial neoplasia (CIN3) (Figure 2) [19], it is likely that L1 antibodies will have reduced efficacy against HPV infected cells which mainly express viral E proteins (Figure 1). Indeed, in women who are HPV positive, prior to vaccination, efficacy has been shown to drop markedly with increasing age at the time of inoculation [34,35]. Thus, it is speculated that additional factors need to be considered with respect to the use of L1 targeting vaccines in HPV-positive women. For example, identification of which HPV types are present combined with the extent of episomal versus integration status, although this is not straightforward. To illustrate, HPV can be present as either episomal or integrated forms, or a mixture of both (Figure 2) [18]. Furthermore, it can also be present as a very low copy number latent infection where the virus is well below the levels needed for detection by most HPV diagnostic systems currently in use [36]. Indeed, the use of a novel, highly sensitive multiplex PCR assay significantly increased detection of multiple HPV types in cervical smears indicating that low copy HPV infection of the cervix may be far more common than anticipated [37]. Thus, HPV infections can embody a complex dynamic system which presents with a wide variety of features. At its simplest, this can be a single HPV type displaying only episomal replication, whereas, at its most complex, infection with mixed-risk HPV types can occur, displaying variable mixtures of latent, episomal or integrated life cycles [38].

## 5. The Influence of HPV Superinfection, Superinfection Exclusion and Latency

As previously discussed, many women are known to acquire cervical infections with multiple HPV types of which only some are associated with cervical cancer (Table 1) [6,9,10]. Since it has been shown that viral E2 proteins from one HPV type can regulate expression from another [28], this implies that the natural history of mixed-type HPV infections may be far more complex than expected. For example, in 1992 a study by Evans et al. noted that women with HPV6/11 anogenital condylomata had a lower prevalence of high-risk HPV-associated CIN lesions [39]. This observation was supported by the work of Silins et al. [40], Luostarinen et al. [41,42] and more recently by Sundström et al. in 2015 [43]. Furthermore, other studies have suggested that HPV types, other than HPV6/11, may also negatively regulate progression to cervical carcinoma induced by high-risk HPV types such as HPV16/18 [44]. It is notable that pairwise co-electroporation of the genomes of HPV18, 31, 39 and 45 into primary keratinocytes showed both positive and negative effects on episomal viral replication depending on which HPV combination was used [45]. This work also demonstrated that HPV45 failed to replicate when paired with any other HPV type and co-expression of types 31 and 39 showed much reduced levels of replication when compared to either virus separately. Furthermore, co-transfection of cells with the genomes of both HVP16 and 18 suppressed replication of both virus types, whereby the E1 helicase from one virus type suppressed genome replication from the other [46]. However, excluding paracrine events, for direct interactions between HPV types to occur, superinfection of single cells with different HPV types is clearly necessary. Further, it is well known that many viruses cannot infect the same cell with two separate types of the same virus. This phenomenon is known as “superinfection exclusion” which has been experimentally demonstrated for HPVs 16 and 18 [47]. Clearly this could provide a rationale for the previously discussed interference between low-risk HPV6/11 and the onset of HPV16-related cervical cancer [39,40,41,42,43]. Furthermore, it has also been reported that individual cells from high-grade CIN3 lesions have only one HPV type present [48] although mixed HPV types have been found in contiguous lesions [49,50]. It is notable that multiple HPV types have been detected in single cells from HPV-positive smear samples, but only in small numbers and when there were no neoplastic changes present [51]. So what is the explanation for these apparently paradoxical observations?

As previously mentioned, it is known that HPV can exist at very low copy numbers, where the virus is present as a non-productive so-called latent infection [36] which has also been confirmed at multiple sites in individual patients [52]. Thus, it is possible that superinfection with different HPV types could occur where one of these is present as a low copy number latent infection and it is not clear how such mixed-type HPV infections would contribute to the development of cervical cancer. Indeed mathematical models of the impact of viral latency on vaccine efficacy have been constructed which demonstrate this would reduce their efficacy by ~25% [53].

These observations illustrate there is a distinct lack of knowledge concerning potential interactions within HPV meta-communities which may be present at various stages of HPV infection and how these may contribute to either progression or regression of disease [54]. In this regard, it has been speculated that differences in infectivity between low-risk and high-risk HPV types combined with sexual promiscuity and the use of L1 targeted vaccines, may actually co-operate to drive the evolution of novel high-risk HPV types [55]. As discussed, the evolution of high-grade CIN generally selects for a single HPV type per cell [48,49,50] although there is some limited evidence for intracellular interactions between different HPV types in normal cervical epithelium [51]. If infection with different HPV types does occur in the same cell, this could either activate or suppress the replication of either virus depending on the HPV types present [45]. However, it is significant that even the simplest explanation of superinfection exclusion provides a potential rationale for why infection with low-risk HPV6/11 may protect against infection with high-risk HPV and the subsequent development of associated neoplasia [39,40,41,42,43]. In view of these observations, given the highly complex nature of HPV biology, it is very clear that administering L1 targeting vaccines, such as Cervarix or Gardasil, to HPV-positive women is not completely without risk, since this could potentially induce unforeseen iatrogenic outcomes [54]. To illustrate, Gardasil prevents infection with HPV16/18/6 and 11 and yet the aforementioned studies indicate that types 6 and 11 may be protective against cervical cancer. The same is true for Gardasil 9, which prevents infection with HPV16/18/31/33/45/52/58/6/11 although its greater L1 target diversity could potentially enhance this effect. For example, cross-type protection could theoretically prevent infection with other low-risk types of HPV the presence of which may act to suppress neoplastic changes as has been demonstrated for HPV6/11 [39,40,41,42,43]. Indeed, it has been previously speculated that within-host interactions between different HPV types could combine with the effects of L1 targeted vaccines to drive the evolution of virulence [56,57], but is there any evidence to support this?

## 6. Prevalence of HPV Types and CIN in Vaccinated Populations

To date, long-term efficacy studies with Gardasil 9 are not available so only Cervarix and Gardasil will be considered further. Prior to, and immediately following approval, reports on the efficacy of Gardasil were produced by the Food and Drug Administration [58] and the European Medicines Agency [59] which both contain results from phase 2 (p005 and p007) and phase 3 Females United to Unilaterally Reduce Endo/Ecto-cervical Disease) trials (p013 and p015, Future I and Future II). An overview of these trials and the different study populations are shown in Appendix A (obtained from [59]).

Analysis of the combined results from protocols p007, 013 and 15 showed that Gardasil was 98–100% effective at preventing HPV6/11/16/18-related CIN2/3 in PPE and MITT-2 women who were sexually naive and HPV negative with normal smears at baseline. However, this dropped to 36.3% in the MITT-3 cohort which represents a real-world population of 16–26-year-old women who were positive for either vaccine or non-vaccine HPV types and also included women who had an abnormal pap test at baseline (Table 257, p341 [58]). Notably, when CIN1, 2 and 3related to infection with non-vaccine HPV types were included, Gardasil showed markedly reduced efficacy in MITT-2 women and provided little benefit (11–13% efficacy) in MITT-3 women (see Table 272, p355 in [58]).

Table 2 (obtained from [58] p. 360) shows the combined results of phase 2 and 3 trials of Gardasil in MITT-3 women stratified by HPV status using both PCR and serology. Gardasil was found to be 98.8–100% effective at preventing HPV16/18-related CIN2/3 in women who were PCR negative for HPV16/18 irrespective of antibody status at baseline. However, in women who were HPV16/18 PCR positive, but seronegative prior to vaccination, the efficacy for preventing HPV16/18-related CIN2/3 was reduced to 31.2%. Most notably, women who were HPV16/18 positive by both PCR and serum antibody test at the time of vaccination showed a negative efficacy of −25.9% on the development of CIN2/3 compared to the placebo control group. Curiously, although it did not reach statistical significance, a moderate improvement in vaccine efficacy from −25.9 to −11.7% (95% CI < 0.0–20.6%) was observed in women who were PCR and serum antibody test positive for HPV6/11/16/18 at baseline (see Table 279, p362 in [58]). Overall these results clearly demonstrate that a positive PCR test for vaccine-covered HPV types at baseline was the most significant factor in predicting reduced vaccine efficacy against CIN and this was reduced still further when combined with a positive HPV serum test. In addition, these data also suggest that a positive baseline HPV6/11 test may be associated with a modest improvement in vaccine efficacy.

The Future 1 p013 study was also analysed for Gardasil efficacy against all grades of CIN related to any HPV type in the baseline HPV-negative RMITT-2 population which includes women who were either sexually naïve or experienced. This demonstrated 31% efficacy where 6.4% (92/1429) of vaccinated and 9.1% (132/1441) of placebo-treated women developed CIN lesions due to any HPV type including types 16/18 (Table 6, p11 [59]). Thus, the overall disease burden, in vaccinated women caused by any HPV type acquired post-vaccination, was ~30% (6.4/9.1) less than the placebo group. Since numerous studies have established that approximately 60–70% of all CIN2/3 and cervical cancers worldwide are caused by HPV16/18 [60,61], this implies that the observed efficacy of Gardasil in baseline HPV-negative women is less than would be anticipated.

A combined analysis of both Future 1 and 2 trials (p013 and p015) was also carried out for CIN2/3 lesions related to any HPV type including types 16/18. This showed a vaccination efficacy of 46% in RMITT-2 and 27% in MITT-2 women against CIN2/3 (Table 15, p22 [59]). Significantly, these data were also stratified for non-vaccine-covered HPV types which showed that >70% of CIN2/3 cases in placebo-treated MITT-2 and RMITT2 women were related to infection with HPV types other than HPV 6/11/16/18. Interestingly, virtually 100% of CIN2/3 lesions detected in vaccinated MITT-2 and RMITT2 women, were related to non-vaccine-covered HPV types, whereas this dropped to 78% in vaccinated and 70% in placebo-treated MITT-3 women. As discussed, this is still much greater than the generally accepted figure of ~30–40% for non-HPV16/18-related CIN2/3 [60,61]. It was speculated this could be due to inaccuracies with the HPV testing methods used and yet it was also stated that the incidence of CIN2/3 increased substantially over time in vaccinated women indicating an important role for non-vaccine HPV types.

The previously discussed FDA and EMA reports on Gardasil were made between 2006 and 2008 which both indicate that the vaccine provides virtually 100% protection against all grades of CIN related to HPV infection with vaccine-covered types in HPV PCR-negative women and many additional studies have since confirmed this observation. However, important questions still remain such as:Extent of vaccine efficacy in HPV-positive women.Extent of HPV cross-type protection.Extent of HPV type replacement and the potential role played by latent infections.

Attempts to address these issues have relied on monitoring pre- and post-vaccination levels of both vaccine-covered and non-vaccine-covered HPV types in various populations. In this regard, a meta-analysis of 9 studies involving 13,886 girls and women ≤19 years of age and 23,340 women 20–24 years of age was carried out by Mesher et al. in 2016 [62]. The results showed that vaccinated women from the younger age group had evidence of cross-protection against HPV 31 although there was also an increase in the prevalence of HPV types 39, 52, 53, 73. However, due to inconsistencies between vaccine types used and age groups it was concluded there was no clear evidence for type replacement although further monitoring was deemed to be important.

The issue of HPV type replacement is still a subject of significant controversy and a study by Gray et al. evaluated pre- and post-vaccination HPV type prevalence in Finnish women [63]. It was concluded that, whilst there was no definitive evidence for type replacement, further study of types 39 and 51 was warranted. A subsequent study by the same group reinforced this conclusion by showing that higher levels of HPV types 51 and 52 were found in more promiscuous vaccinated versus unvaccinated women [64]. It is significant that a more recent study concluded it is still too early to eliminate the possibility of type replacement [65]. It is also noteworthy that many of the studies on HPV type replacement have examined virus types in vaccinated versus unvaccinated women with normal cytology and there are much fewer studies that have evaluated this in CIN or cancers. In this regard McClung et al. evaluated the prevalence of CIN2 and associated HPV types in American women pre- and post-vaccination with Cervarix or Gardasil between 2008 and 2016 [66]. A very significant reduction in the prevalence of CIN2 was observed in women aged 18–19 and 20–24 years although there was also a pronounced trend towards increasing prevalence of HPV types not covered by the vaccines in vaccinated women aged > 24 years (Figure 3, adapted from [66]).

Furthermore, it was shown that a substantial proportion of this increase was due to increased prevalence of additional HPV types covered by Gardasil 9. However, there was also an overall increase of ~12% in the prevalence of CIN2 in presumably catch-up vaccinated women aged between 30 and 35 years which was attributed to changes in screening practices between 2008 and 2016.

A similar study was carried out in New Zealand which analysed changes in HPV types associated with CIN2 between 2013 and 2016 in 392 women under the age of 25 [67]. New Zealand commenced vaccination of women aged > 18 years with Gardasil in 2008 which was extended to girls/women aged between 9 and 20 years in 2009. It can be seen that, between 2013 and 2016, there was a pronounced shift in the prevalence of HPV types detected in CIN2 lesions (Figure 4, adapted from [67]).

Although HPV16/18 lesions decreased substantially, there was a 33% increase in lesions associated with other high-risk types and it was noted there was a significant increase in the number of women diagnosed with high-grade cytology over the same period (see Supplementary Figure mmc1, Innes et al. [67]). Interestingly, unvaccinated women also showed a very pronounced drop from 66% HPV16/18-positive CIN2 lesions in 2013 to 17% in 2016. This was attributed to reduced sexual transmission of the virus via herd effects since increased numbers of vaccinated immune women should reduce the chances of sexual transmission to unvaccinated women.

More recently, Innes at al evaluated the effect of vaccination on the incidence of abnormal cervical cytology and histology in 104,313 New Zealand women aged 20–24 years between 2010 and 2015 [68]. Compared to unvaccinated women, the incidence of high-grade CIN2+ decreased 30% in women vaccinated at <18 year of age but only a 7% decrease was found in women vaccinated at >18 years. Furthermore, there was a more modest effect on low-grade lesions which were 15% lower in women vaccinated at <18 years, whereas women vaccinated at >18 years actually showed a 7% increase in incidence. Since it is generally accepted that approximately 60% of high-grade cervical lesions are HPV16 and/or HPV 18 positive [60,61] the expectation was for a more marked effect. However, although previous studies in New Zealand have shown that 62% of CIN3 lesions are positive for HPV16 and/or HPV18, 31% have additional high-risk types present [69] which implies these may have subsequently assumed dominance to drive the development of CIN.

Drolet et al. carried a systematic review and meta-analysis on the population level impact and herd effects related to HPV vaccination derived from 65 articles covering 60 million individuals over a maximum period of 8 years in order to compare pre- and post-vaccination relative risk of HPV-related end points [70]. This study clearly supports herd effects demonstrating the impact of vaccination on HPV prevalence and CIN2+ in both girls and women in addition to effects on the incidence of anogenital warts in girls, women, boys and men. The data shown in Appendix A in Drolet et al. [70] show that vaccination reduced the prevalence of HPV16/18 in girls aged 16–19 years although there was an increase in the prevalence of non-vaccine high-risk HPV types which was also observed in women aged from 20 to 24 years and 25 to 29 years (Appendix A Drolet et al. [70]). It was concluded that this was due to either unmasking of HPV16/18 or, alternatively, HPV type replacement. However, it is significant that women in the 25–29 year age groups were mostly unvaccinated and yet the post-vaccination relative risk of low-risk HPV-associated genital warts declined in countries with high vaccine coverage in both sexes and all age groups—presumably due to reduced virus transmission via herd effects. However, the same trend was not observed for CIN2+ where the incidence declined in the 16–19 and 20–24 age groups but actually increased by ~20% in women older than 24 years. It was speculated this increase could be related to changes in screening recommendations, sexual practices or health-seeking behaviour and yet it is curious that the risk of genital warts declined significantly in older women, presumably due to vaccine-related herd effects. This raises the question, why did this herd effect not reduce, or at least stabilise, the risk of CIN2+ in older women? It is very possible this could be related to differences in development time between these two pathologies. However, an alternative explanation could be the quadrivalent vaccines ability to prevent infection with low-risk types of HPV since this may actually promote the development of neoplastic changes as previously discussed [39,40,41,42,43]. If this hypothesis is correct, it would be predicted that bivalent vaccines, which do not target low-risk types of HPV, may be more effective in older women but is there any evidence to support this?

Unfortunately, there are currently no studies with sufficient follow up and suitable efficacy end points to definitively answer this question although it is known that the bivalent vaccine provides 91% efficacy at protecting women older than 25 which lasts for 7 years [71]. Furthermore, analysis of multiple phase III studies showed that the efficacy of the bivalent vaccine against CIN3+ caused by any HPV types, was 92%, whereas the quadrivalent vaccine was only 43%. If this is further restricted to CIN3 not caused by HPV16/18, the bivalent vaccine still has >80% efficacy, whereas the quadrivalent vaccine has negligible or negative efficacy [72]. It is speculated this could be due to differences between the adjuvants used in the different vaccines although an alternative explanation is that the quadrivalent vaccine causes a more pronounced disturbance in the balance between high-risk and low-risk HPVs present in genital epithelium. Thus, a combination of direct effects in younger women and indirect transmission-related herd effects in older women could cooperate to produce increased risk of CIN3.

It is very clear that the previously discussed trial results raise many questions and do not provide conclusive evidence that any of the prophylactic vaccines will actually prevent cervical cancer. Indeed a very recent review by Rees et al. critically assessed the available published data in relation to this goal and drew the same conclusion [73]. These authors provided the following recommendations for future vaccine clinical trials:Vaccinate prior to onset of sexual activity and begin assessment of end points at age of usual cervical screening once sexually active.Make all clinical study reports, including anonymised individual patient data, publicly available.Separate trials to assess benefits in women already exposed to HPV without restrictions based on risk factors.Analyse data by country and study site.Ensure the testing interval is in line with usual cervical screening protocols.Continue follow up for a minimum of 20 years from time of sexual debut.Power trials for primary composite outcomes CIN3/AIS/cervical cancer due to oncogenic HPV types.Define secondary outcome of persistent HPV16/18 infection at a minimum of 12 months.Use standardised methods for HPV testing.Undertake a saline placebo-controlled efficacy trial of Gardasil 9 in previously unvaccinated populations as it is difficult to draw conclusions on efficacy and risk of harms based on comparing Gardasil 9 to Gardasil.

Although implementation of these suggestions would obviously clarify some of the issues raised, it is very clear that such trials would also benefit from improved HPV testing in relation to vaccine efficacy. For example, analysis of the prevalence of all 51 genital HPV types [9] in women vaccinated with either the bivalent, quadrivalent or nonavalent vaccines would provide much needed information on the potential impact of these products on the balance between high- and low-risk types of HPV in genital epithelium. In addition, correlation of these results with respect to cervical cytology/pathology, and ultimately cancer, could also shed light on any cause and effect relationships which may exist, although this is not straightforward. For example, many studies draw conclusions on vaccine efficacy by comparing unvaccinated older women to vaccinated younger women at the same geographical location and time period. However, these may not be accurate since herd effects have been shown to affect the balance between HPV types present in both vaccinated and unvaccinated women in the same population [64,67,70,74].

## 7. Incidence of Cervical Cancer in Vaccinated Populations

Selection of the most appropriate outcome is critical for assessing vaccine efficacy and CIN3 was judged to be preferable to either HPV status or low-grade disease since these have much higher rates of spontaneous regression [73]. However, cervical cancer incidence is clearly the best indicator of vaccine efficacy although it can be argued it may be too early for the vaccines to have had any appreciable impact on this. Nevertheless, since most vaccination programmes started between 2006 and 2008, it may be informative to assess changes in the incidence of cervical cancer rates in vaccinated populations as near to the present day as possible. In this regard, it is notable that several countries with extensive vaccination programs have since noted an increased incidence of cervical cancer in 2018 [75].

In the UK the overall incidence of cervical cancer decreased by ~35% from 1990 to 2005 and vaccination was started in 2008.

The overall incidence of cervical cancer remained at a constant level until 2017 (Figure 5A). However, a more detailed analysis of these results stratified by age groups showed there was a significant increase in incidence in the 25–34 year age group which was less apparent in the 35–49 year age groups (Figure 5B, obtained from Cancer Research UK website [76]). It was concluded that this could be the result of changes in screening programmes coupled with more sensitive HPV detection methods. In the UK girls aged 12–13 years were vaccinated with additional catch-up vaccination offered to girls aged 14–18 years which achieved a coverage of >86%. Thus, it is clear that catch up-vaccinated girls aged >16 years at the time of vaccination will be included in the 25–34 years age group which showed the largest increase in incidence of cervical cancer in 2017. Clearly the vast majority of women aged 35–49 years will not have been vaccinated although, as discussed, this does not exclude indirect vaccine-related herd effects on HPV transmission.

Australia started Gardasil vaccination in 2007, achieving a coverage of >80%. Figure 6 shows the age-stratified incidence of cervical cancer over time which was obtained from Cancer Australia’s National Cancer Control Indicators website [77].

The age-stratified five-year average cervical cancer incidence in New Zealand from 1987 to 2016 is shown in Figure 7 (obtained from Clinical Practice Guidelines for Cervical Screening in New Zealand, 2020 [78]). It can be seen that, after introduction of the National Cervical Screening Program from 1990 until 2007, there was a ~2.5-fold drop in incidence in all age groups except for women aged 20–24 years where the incidence of disease is very low anyway. It is curious that, following the introduction of vaccination in 2008/2009 until 2016, there was no further reduction in incidence in any age group apart from women aged 50–69 years. Indeed, women aged between 25 and 49 years actually showed a modest increase and vaccine coverage in New Zealand at this time was estimated to be ~50%. Once again, vaccine-related herd effects on HPV transmission cannot be excluded from the 25–49-year-old cohort.

Sweden introduced cervical screening in the mid-1960s and by 2011 the incidence of cervical cancer had declined 3 fold. Vaccination started in Sweden in 2008 firstly with Cervarix and then Gardasil from 2012 with a coverage of 10% in the period 2008–2010 rising to >80% in 2018. Figure 8 (obtained as supplementary data from [79]) illustrates the incidence of cervical cancer in Sweden between 1960 and 2017 where it can be seen there was an overall increase of 20% from 2014 to 2017. Wang et al. further analysed this effect in women aged 29–65 years and demonstrated an overall increase of 59% for all age groups between 2014 and 2015 when compared to that observed between 2003 and 2013 [79]. However, it is significant that the largest increase of 96% was seen in the 29–39 year age group and it was noted that this effect was only observed in women who attended for regular screening and not in unscreened women, although the latter had a 6-fold higher incidence at baseline.

The conclusion from this study was that the increase may be due to insufficient detection of precursor lesions prior to 2009 and the possibility that vaccination may have played a role was dismissed since <0.5% of women in the study cohorts were vaccinated. Indeed a more recent study of vaccination and the risk of cervical cancer in Sweden concluded that vaccinated women aged 30 years or less had a significantly lower risk than unvaccinated women [80]. However, as previously discussed, many studies have now reported vaccine-related herd effects on the transmission of both high- and low-risk HPV types to unvaccinated women, which indicates this could still play a role in the observed increased incidence of cervical cancer. Indeed another Swedish study analysed the effects of vaccination on the prevalence of 27 HPV types in cervical smears from 15 to 23-year-old Swedish women from 2008 to 2018 [74]. A significant reduction in vaccine-covered HPV16 was observed in both vaccinated and unvaccinated women although there was also a substantial 2–5-fold increase in high-risk HPV types 56, 51, 59, 32 and 39, which are not included in Gardasil 9 (see Figure 6 in Ahrlund-Richter et al. [74]). Furthermore, there was also a significant reduction in the incidence of non-vaccine-covered low-risk HPV types 43 and 44 in vaccinated versus unvaccinated women in the 2017–2018 cohort. Collectively these results indicate that vaccine-related herd effects on HPV transmission may increase infection with non-vaccine high-risk types and reduce infection with non-vaccine low-risk types. Limitations of this study are its small size since a total of only 1131 cervical smears were analysed although robust statistical outcomes were deduced. Restricting the age to 15–23 years was also a limitation as the potential influence of herd effects in older women would have been more informative since they have a higher incidence of CIN3 and cervical cancer. However, that being said, this work clearly supports HPV type replacement in vaccinated and unvaccinated women which implies that direct or indirect (herd) effects of the vaccines on the prevalence of HPV types could still be a factor in the recent increase in the incidence of cervical cancer in Sweden and elsewhere.

Norway started single cohort vaccination of 12-year-old girls with Gardasil in 2009 achieving a coverage of 89% but, unlike most other nations, did not start catch-up vaccinations until 2016 [81]. It is notable that Norway also observed an increase in the incidence of cervical cancer since vaccination started [82] which is consistent with an increase in the CIN2/3 observed over the same time period [81].

It is significant that the UK, Australia, New Zealand, Sweden and Norway all have vaccination coverage rates of 50–86%, whereas France started vaccination in 2007 and only achieved coverage of 19% in 16-year-old girls. Indeed France has the lowest coverage in Western Europe which begs the question, how does the incidence of cervical cancer in France compare to nations with higher vaccine coverage over the same period? In 2007, the age-standardised incidence of cervical cancer was ~10 per 100,000 women, whereas in 2018 it had declined to 6.7 [75]. Thus, unlike other nations with higher vaccine coverage, the incidence of cervical cancer in France has continued to decline post-vaccination.

## 8. Discussion and Conclusions

Clearly the preceding observations do not prove any causal relationship between the current prophylactic vaccines and iatrogenic outcomes with respect to cervical cancer. However, it is also very clear that there is a distinct lack of knowledge concerning the complex biology of mucosal HPV communities and the purpose of this review was to draw attention to how this could be affected by L1 targeting prophylactic vaccines. With regard to assessing vaccine efficacy, the recommendations of Rees et al. [73] are clearly very pertinent. However, in view of the additional observations made herein, it is proposed that the following suggestions should also be considered:Use of high-sensitivity methods of HPV analysis in order to detect latent infections.Determination of the impact of vaccination over time on the prevalence of all 51 HPV types known to infect genital mucosa in all age groups and combine this with vaccination status, age at vaccination, vaccine type and sexual history.Use of this to discriminate between direct and indirect (herd) effects of vaccination on the incidence of different HPV types in unvaccinated women.Carry out these analyses on normal, CIN1, 2 and 3 and cervical cancers.Do not assume that unvaccinated women residing in a vaccinated population will experience only beneficial herd effects since increased transmission of non-vaccine-covered high-risk HPV types combined with decreased transmission of low-risk types has been shown to occur [74].Conclusions on vaccine efficacy should not be based on comparing vaccinated and unvaccinated women over time at the same geographical location since herd-effect transmission may confound this.Ideally, the incidence and disease association of different HPV types should be evaluated in any population before and after implementing vaccination.

Whilst it is very clear that the current HPV vaccines have substantial efficacy against disease related to vaccine-covered HPV types, there are many questions left unanswered concerning the full spectrum of their effects on HPV biology. As discussed, HPVs of the genital mucosa can represent a dynamic meta-community which changes over time with respect to many factors such as age, promiscuity and type of contraception used which means it is difficult to predict outcomes. In this regard, it is notable that the French PAPCLEAR trial [83] has some of the improvements suggested herein and the first results of this are now available as a preprint [84]. Limitations of this study are its small size (149 participants), with no possibility of stratification with respect to pathology/cytology. Interestingly, vaccinated women only showed a reduction in the prevalence of high-risk HPV16, whereas non-vaccine-covered high-risk types 31, 51 and 59 were also substantially reduced. Furthermore, the incidence of non-vaccine-covered high-risk types 52 and 59 increased in vaccinated women, whereas low-risk types 66 and 54 were reduced. However, as indicated, these results will undoubtedly be subject to herd effects on HPV transmission, which means it may not be possible to delineate the effects of vaccination between vaccinated and unvaccinated women at the same geographical location. Although not exhaustive, Figure 9 illustrates some of the potential HPV-related interactions which may occur between vaccinated and unvaccinated women.

Finally, as discussed, the recent increase in the incidence of cervical cancer in several countries with high vaccine coverage does not in any way establish cause and effect and there have been a variety of explanations put forward to explain this, such as:Changes in sexual practice.Changes in screening practice.Problems with assay sensitivity.Influx of migrant communities.

Indeed it has been speculated that this may be due to relaxed screening intervals which were adopted by some countries in order to cover the cost of vaccination and HPV testing. However, it is very unlikely that these same factors will be present in all the countries cited and it is very clear it will be of crucial importance to analysed the incidence of HPV types, CINs 1–3 and cervical cancers in women of all age groups in countries with high and low vaccine coverage in the next decade.

## Figures and Tables

**Figure 1 viruses-13-00022-f001:**
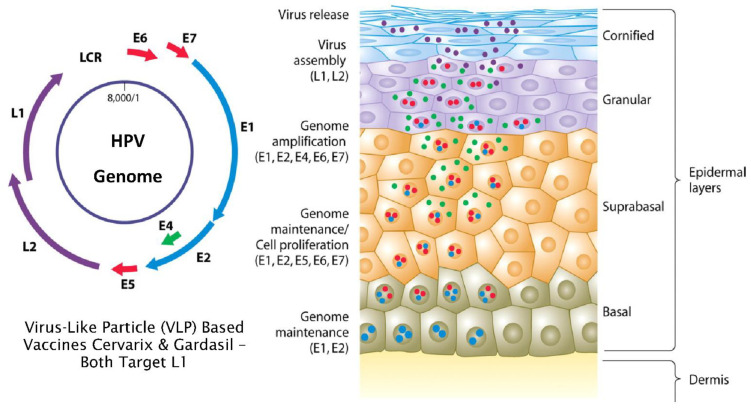
Normal full HPV life cycle.

**Figure 2 viruses-13-00022-f002:**
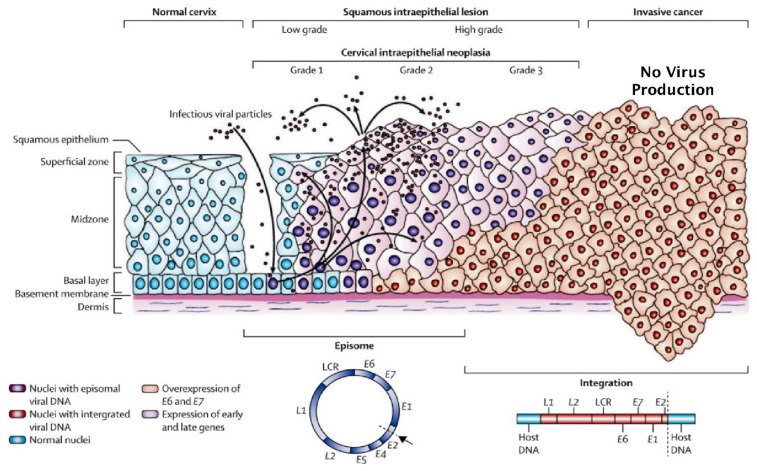
Truncated HPV life cycle and neoplasia. The E2 ORF is disrupted by integration into host DNA, which prevents the production of infectious virions, upregulates expression E6 and E7 and suppresses expression of L1.

**Figure 3 viruses-13-00022-f003:**
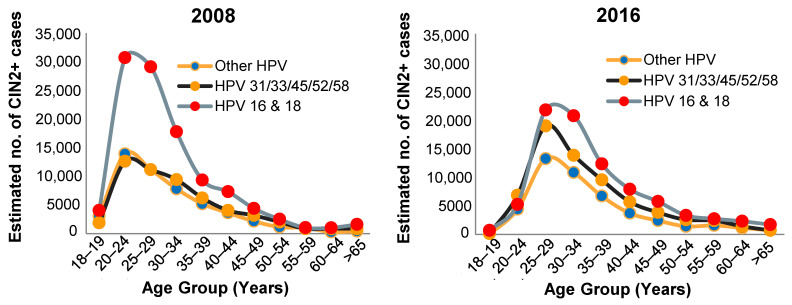
Estimated number of diagnosed CIN2+ cases by HPV type and age group in United States from 2008 to 2016.

**Figure 4 viruses-13-00022-f004:**
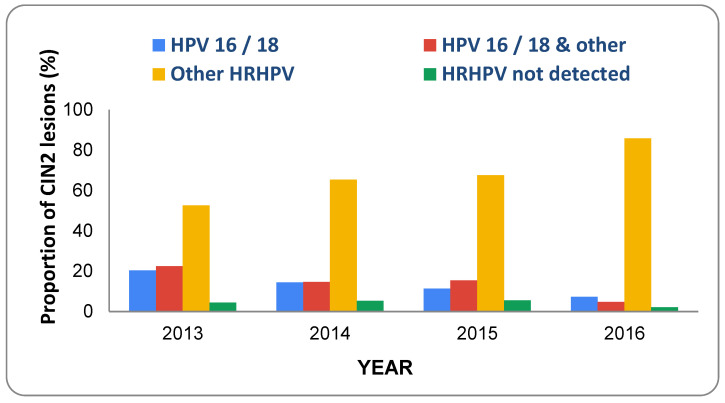
CIN 2-associated HPV types detected in New Zealand women from 2013 to 2016.

**Figure 5 viruses-13-00022-f005:**
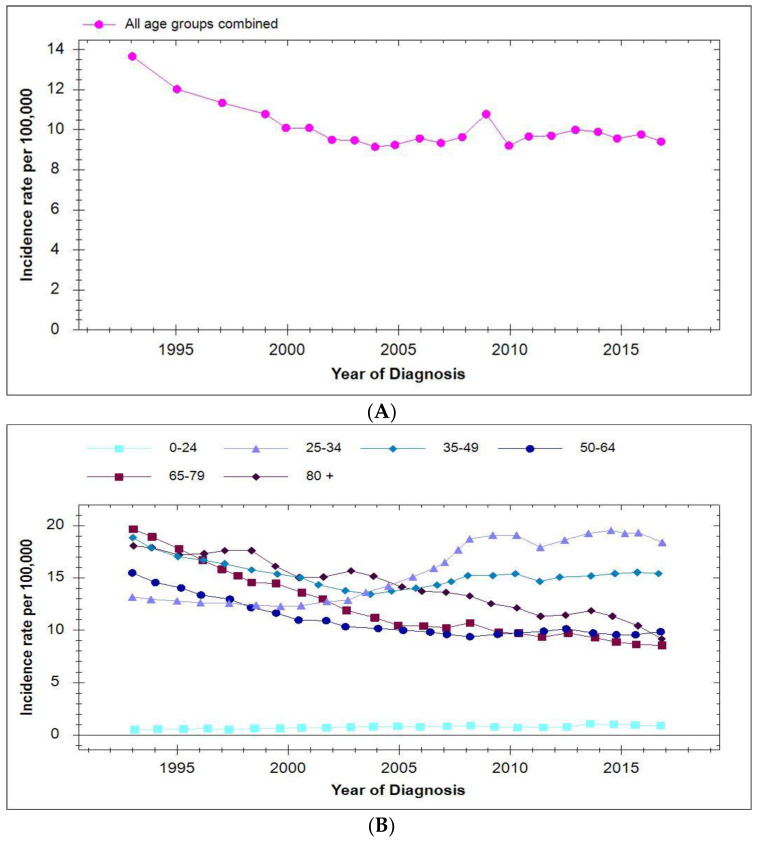
Incidence of invasive cervical cancer in the UK 1993–2017. (Source: Cancer Research UK.) (**A**) Total for all age groups; (**B**) Stratified between different age groups.

**Figure 6 viruses-13-00022-f006:**
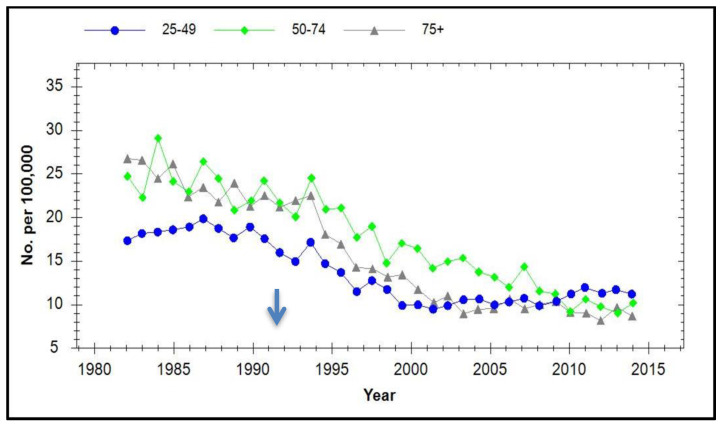
Age stratified incidence of cervical cancer in Australia from 1982 to 2015. Since the introduction of cervical screening in 1991, the incidence in the 25–49 years age range declined until 2001 then started to increase from 2009 until 2015. It is speculated that restructuring these results into the age groupings used for the UK data, as shown in Figure 5B, may prove informative.

**Figure 7 viruses-13-00022-f007:**
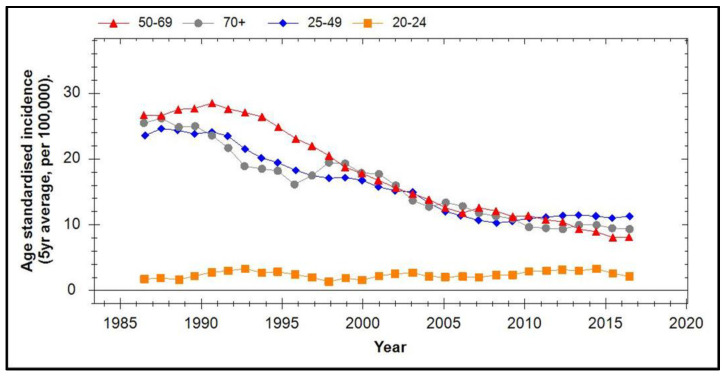
Five-year averaged age-standardised incidence of cervical cancer in New Zealand from 1987 to 2017.

**Figure 8 viruses-13-00022-f008:**
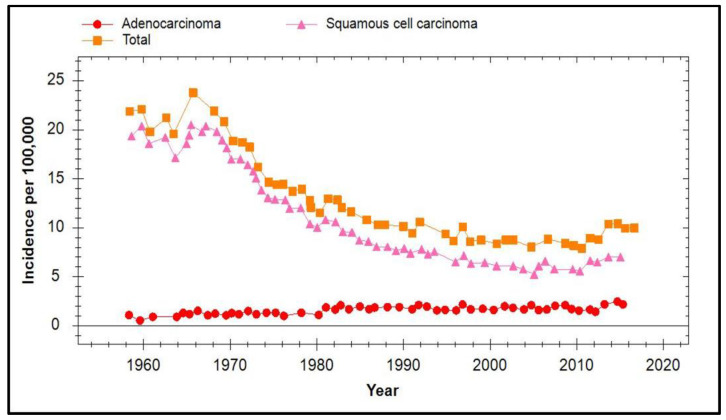
Incidence of cervical cancer in Sweden from 1960 to 2017.

**Figure 9 viruses-13-00022-f009:**
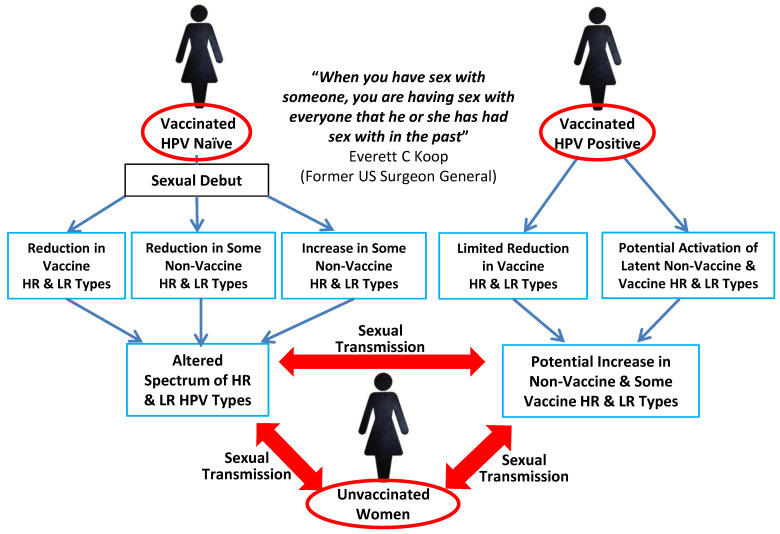
Potential for dissemination of vaccine-related changes in HPV types between vaccinated and unvaccinated women. Illustrates how vaccine-induced changes in the prevalence of different HPV types in vaccinated women could be disseminated to unvaccinated women via partner-mediated sexual transmission.

**Table 1 viruses-13-00022-t001:** HPV type and disease association.

Disease	HPV Type
Plantar warts	**1**, **2**, 4, 63
Common warts	**2**, **1**, **7**, 4, 26, 27, 29, 41, 57, 65, 77, 3, 10, 28
Flat warts	**3**, **10**, 26, 27, 28, 38, 41, 49, 75, 76
Other cutaneous lesions (e.g., epidermoid cysts, laryngeal carcinoma)	6, 11, 16, 30, 33, 36, 37, 38, 41, 48, 60, 72, 73
Epidermodysplasia verruciformis	**2**, **3**, **10**, **5**, **8**, **9**, **12**, **14**, **15**, **17**, 19, 20, 21, 22, 23, 24, 25, 36, 37, 38, 47, 50
Recurrent respiratory papillomatosis	**6**, **11**
Focal epithelial hyperplasia of Heck	**13**, **32**
Conjunctival papillomas/carcinomas	**6**, **11**, 16
Condyloma acuminata (genital warts)	**6**, **11**, 30, 42, 43, 45, 51, 54, 55, 70
Cervical intraepithelial neoplasia	
Unspecified	30, 34, 39, 40, 53, 57, 59, 61, 62, 64, 66, 67, 68, 69
Low risk	**6**, **11**, 16, 18, 31, 33, 35, 42, 43, 44, 45, 51, 52, 74
High risk	16, 18, 6, 11, 31, 34, 33, 35, 39, 42, 44, 45, 51, 52, 56, 58, 66
Cervical carcinoma	**16**, **18**, 31, 45, 33, 35, 39, 51, 52, 56, 58, 66, 68, 70

Order indicates relative frequency and bold indicates the most frequent association (Burd [6]).

**Table 2 viruses-13-00022-t002:** Protocols 005, 007, 013, and 015: efficacy against HPV 16/18-related CIN 2/3, AIS or worse—MITT-3 population, by initial baseline HPV status.

Gardasil N = 10268				Placebo N = 10273			
Day 1 Status	N *	No. of Cases	Incidence Rate/100 Person Years at Risk	N *	No. of Cases	Incidence Rate/100 Person Years at Risk	Vaccine Efficacy 95% CI
MITT-3	9831	122	0.6	9896	201	0.9	39.0% (23.3, 51.7%)
PCR (-) Sero (-)	9342	1	0.0	9400	81	0.4	98.8% (92.9, 100.0%)
PCR (-) Sero (+)	853	0	0.0	910	4	0.2	100.0% (−63.6, 100.0%)
PCR (+) Sero (-)	661	42	3.2	626	57	4.6	31.2% (−4.5, 54.9%)
PCR (+) Sero (+)	473	79	9.1	499	69	7.3	−25.8% (−76.4, 10.1%)
Sero and/or PCR (+)		(121)			(130)		(No efficacy estimate provided)

* Some subjects are counted in more than one row due to different baseline PCR/serostatus for HPV 16 and HPV 18. Each subject is counted once within each applicable row for HPV 16 or HPV 18. N = number of subjects randomised to the respective vaccination group who received at least one injection. n = Number of subjects evaluable. () = total number of cases where subjects are PCR + and/or sero+ in the respective group. Source: p. 360 in https://www.impfkritik.de/download/gardasil_fda_464_pages.pdf.

## Data Availability

No new data were created or analyzed in this study. Data sharing is not applicable to this article.

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
