# Peer review of "Potential Effects of Human Papillomavirus Type Substitution, Superinfection Exclusion and Latency on the Efficacy of the Current L1 Prophylactic Vaccines"

_viruses, 2020, doi:10.3390/v13010022_

Round 1

Reviewer 1 Report

The authors have addressed all of the raised concerns

Author Response

We thank the reviewer for their constructive criticisms which we have acted on.

Reviewer 2 Report

Overall, I found the review easy and interesting to read, with proper references. It touches several issues we had long discussions with my former PI (especially type replacement and herd immunity). Also, I feel a major issue is relaxed screening intervals many countries chose to cover the costs of vaccination and HPV testing. Maybe worth highlighting more.

There are some minor punctuation and typing corrections, but I feel only minor revisions are required. 

In detail:

Line 42: Table 1: In line two you have included types 1 and 4 twice.

Line 43: Maybe add also the bibliography number after "(Burde)".

Line 46: "found associated" Rephrasing would make it more to the point.

Line 96: The reference 20. Cheung, J. L. et al, mainly focuses in episomal HPV related lesions. Perhaps rephrasing or removing the citation, or transferring it with citation no 22

Line 108: Why can't I find the referenced document???

Line 130: Rephrasing "acquisition of HPV infections"

Lines 132-133: I feel the efficiency of vaccination to protect from HGSIL and CAs (shown in many studies) is underestimated with this phrasing. 

Line 145: Please check for double space before "efficacy"

Line 164: Please check if a space is present prior to "anogenital"

Line 167: Correct "SundstrOm"

Line 185: "were present" instead of "are present"

Line 208: Should a comma be present prior to "since"?

Line 220: Cannot find the cited document [58]

Line 221: Check if there are two spaces present after "p007)"

Line 236: Why is a FDA document referenced from a German forum address?

Line 282: Why o you have a dash after ":". The same goes for line 512.

Line 284: End of sentence with ";".

Line 387: Missing full stop.

Line 405: Missing commas?

Line 425: Two periods present. Same goes for line 548.

Line 441: Space present after "1982".

Line 497: Full stop missing.

Line 503: Change "declined" to "declined to"?

Author Response

This manuscript is a resubmission of an earlier submission. The following is a list of the peer review reports and author responses from that submission.

Round 1

Reviewer 1 Report

The review from Ian N Hampson et al described HPV types replacement, exclusion, latency and their potential influence on HPV vaccines. This is a valuable and special view to discuss the current vaccine efficiency. However, there are some main concerns about the manuscript I would like to discuss.

  1. The current vaccination is against up to 9 types of HPV types, which showed significant effects on reducing lesions and cancers caused by these HPV types. As the vaccines are designed, they are not meant to target all the HPV types. So, if the vaccine related HPV types are blocked by vaccines, other non-vaccines types will show up, especially when lots of data are presented as percentage. But this is not necessarily due to vaccination itself. It is also debatable if this can be called “replacement” or “substitution”. The non-vaccine types have also been observed increasing among unvaccinated women.

Many potential reasons can cause higher percentage of other types, such as sexual behavior changes in modern days, which will lead to HPV prevalence increasing in both vaccine and unvaccinated group. Other reasons such as the detection method sensitivity or competition without HPV 16 and 18 in the samples and methods could also contribute to this change. The authors only mentioned very little about other possibilities and put much more effort to claim the HPV substitution due to the vaccination. Besides, in Fig 3, other HPV did not show a clear increase as the authors claimed between lines 309 to 311.

  1. Authors tended to mention HPV vaccination efficiency just in one or two sentences, while listing out their uncertain hypothesis/questions several times. As a reader, it is quite misleading and easy to conclude that it is the vaccine that caused cervical cancer, which is not supported or proved by current studies. I strongly suggest authors stand in an unbiased view and also discuss the vaccine efficacy for lesions and cancers caused by HPV 16 and 18 as well. For example, lines 281-282, authors focused especially on -2.0%, which is probably not significant from the value of 95%CI, and claimed the vaccines have no benefit against CIN3. However, this section in the table is for HPV not covered in the vaccines. The vaccines are not expected to have the function to all HPV types from the beginning, so it is not surprising that it does not inhibit other HPV types. Similarly, one should not expect the HBV vaccine has function on cervical cancer. So, it is not fair to use the data on other HPV types to argue the current HPV vaccines do not have enough efficacy.

  1. The review article should base on the facts of published studies, not further or over-interpreted the result. Lots of current publication on the HPV vaccine is only presented as “correlation”, but authors use these observations and draw hypothesis as “causality”.

For example, references 28-29, all the studies just observed less frequency of HPV 16, 18 related cervical lesions from the participates with low-risk HPV infection before. However, this does not necessarily mean HPV low-risk types inhibit the high-risk types infection, and remove low-risk types from vaccines will potentially benefit patients later as authors hinted. There are other possibilities such as the human antibody against low-risk types after infection, helps to against high-risk related infection and diseases. In this case, having low-risk HPV infection will protect later high-risk HPV infection, but removing low-risk HPV from current vaccines will not benefit later infection.

Without solid prove, one can have so many interpretations and hypotheses, but this should not be a commentary or opinion paper instead of a review article.

  1. Another concern is the herd immunity authors talked many times in this review. Again, there is no solid proof for herd immunity, even in the high vaccinated coverage countries. Probably there is a certain level of protection observed in those countries, such as presented in ref 59. However, giving only one gender got vaccinated and the vaccine coverage is low among women before the vaccination program. It is still debatable if herd immunity is achieved as authors claimed in this review. The published papers did not draw conclusions on herd immunity achievement, so authors should not interpret the data from these papers to fit their hypothesis.

  1. Vaccination time and cervical cancer cases discussion for figures 5-8: The authors claimed there is no big drop as expected in cancer cases after the vaccination, but instead a certain increase as in Fig 8 can be observed. This strongly hints for the reader that the vaccination itself caused the increase of cancer. However, the vaccine program is targeting teenagers, while the cancer patients are not the same group of people at all. It is misleading to draw a vaccination arrow on the adapted figures giving the group of people are different. In addition, a recently published large cohort paper on cervical cancer from Sweden showed HPV vaccination was associated with a substantially reduced risk of cervical cancer (DOI: 10.1056/NEJMoa1917338).

  1. Figure 9 is also misleading, as I mentioned above, the vaccine aims to target the types covered in the vaccine. The situation authors present could happen, but it is definitely not the whole picture. It is just the authors own hypothesis. The result if this hypothesis holds is that women will soon need new HPV vaccines that target other types. But the current vaccine against HPV16&18 is still needed and the women who got vaccinated are still vaccinated. They have not become “unvaccinated”. Besides, the non-vaccine types have also been increased among other groups including unvaccinated and pre-HPV positive women, which has not been presented in this figure.

  1. Lastly, the review is very long with lots of tables and figures from other publications. Lots of them could be removed since they are abundant together with the text. Have all the figures and tables got copyright from the published journals? It is also not the same fond in the figures throughout the manuscript. I suggest authors shorten the text from section 6, focus on facts and data presented in the published papers, without own interpretation. Please keep the discussion for all the papers concisely. Also, discuss the vaccination efficiency first before talking about the potential risk. Please do not make too many conclusions/hypotheses/questions on uncertain possibilities. Stress the uncertainty, so the information does not mislead readers.

Reviewer 2 Report

The manuscript entitled “Potential Effects of Human Papillomavirus Type-Substitution, Superinfection-Exclusion and Latency on the Efficacy of the Current L1 Prophylactic Vaccines” aims to explore in detail the biology of mucosal HPV’s and their potential for interacting with the current prophylactic HPV vaccines, to produce unexpected outcomes.

The main strength of the review is its focus on the relationship between HPV life cycle/oncogenesis and current prophylactic HPV vaccines. However, the vaccine efficacy/effects of only cervarix and Gardasil are mainly described, which is probably due to the lack of extensive studies about the nonavalent vaccine.

With few exceptions, the review covers a vast number of published papers. The most recent papers are quoted. I have suggested some improvement

The English language/style of sufficient quality. Figures/tables are fine.

SPECIFIC  COMMENTS

Line 35 ”…. many others are yet to be identified [6, 7].” --> as example, the recent identification of a novel Beta-2 papillomavirus could be interesting to be reported (PMID: 30834389)

Line 41 “…. For example, type 16 has the highest oncogenic potential” --> As correctly indicated by the authors, HR-HPVs (mainly HPV16/18) has the highest oncogenic potential. In this context, the main HPV-driven tumors should be briefly mentioned. Indeed, penile/anus cancers (PMID: 21543996), vulvar cancer (PMID: 32266002), as well as tumors of the upper respiratory tract, including head and neck cancers (PMID: 32362526), have been found to be driven by oncogenic HPV infection. For completeness of information, the tumors/references mentioned above, should be described in this introductive section.

Line 91. “disruption of the E2 open reading frame when the virus integrates with cellular DNA of the host as illustrated in Figure 2 or, alternatively, DNA methylation of the E2 binding site in the virus long control region (LCR) [14-16].” --> The viral epigenetics plays an important role in this context. E6/E7 expression is mediated by the interaction between E2 and two regulatory elements of the HPV genome, i.e. the L1-LCR 5’ and the promoter [PMID: 17278110]. Since the binding of E2 with these elements depends on their methylation status, E6/E7 expression is also under epigenetic control [PMID: 17278110]. The role of HPV DNA methylation during HPV DNA replication/life cycle should be more deeply described.

Line 125 “…However, for >15 years, there has been considerable debate on the ability of Cervarix and Gardasil” --> For a detailed description of the nonavalent vaccine please see PMID: 32630772

Line 133 “…is some evidence of cross-protection against a few other high-risk types” the cross protection against non-vaccine HPV types has also been evaluated in other studies that deserve attention (PMID: 18499916; PMID: 24553190)

Line 168 “..and it has been demonstrated for HPV’s 16 & 18” --> Additional studies also reported on the interference between multiple HPV types. Indeed, it seems that seropositivities for HPV6/11 prevents the HPV16-related cervical cancer onset [PMID: 10580926, PMID: 10074912].

Line 424 “In the UK the overall incidence of cervical cancer decreased by ~35% from 1990 – 2005 and 424 vaccination was started in 2008.” ---> however, an explanation could be the increase of prevention programs among women in recent years as well as the development of more sensitive HPV detection methods

FIGURES

Fig 1. “genom” --_> genome?
